# trRosettaRNA: automated prediction of RNA 3D structure with transformer network

Wenkai Wang [1,5], Chenjie Feng[2,3,5], Renmin Han[2,5], Ziyi Wang[2], Lisha Ye[1], Zongyang Du[1], Hong Wei[1], Fa Zhang [4] ✉, Zhenling Peng [2] ✉ & Jianyi Yang [2] ✉

RNA 3D structure prediction is a long-standing challenge. Inspired by the recent breakthrough in protein structure prediction, we developed trRosettaRNA, an automated deep learning-based approach to RNA 3D structure prediction. The trRosettaRNA pipeline comprises two major steps: 1D and 2D geometries prediction by a transformer network; and 3D structure folding by energy minimization. Benchmark tests suggest that trRosettaRNA outperforms traditional automated methods. In the blind tests of the 15th Critical Assessment of Structure Prediction (CASP15) and the RNA-Puzzles experiments, the automated trRosettaRNA predictions for the natural RNAs are competitive with the top human predictions. trRosettaRNA also outperforms other deep learning-based methods in CASP15 when measured by the Z-score of the Root-Mean-Square Deviation. Nevertheless, it remains challenging to predict accurate structures for synthetic RNAs with an automated approach. We hope this work could be a good start toward solving the hard problem of RNA structure prediction with deep learning.

Ribonucleic acid (RNA) is one of the most important types of functional molecules in living cells. It is involved in many fundamental biological and cellular processes, for example, as the transcript of genetic information, serving catalytic, scaffolding, and structural functions. Interest in the structure and functions of non-coding RNA (ncRNA), such as transfer RNAs (tRNAs) and ribosomal RNAs (rRNAs), has been increasing over the past few decades with the discovery of new types of ncRNAs every year. Similar to proteins, ncRNA molecules' biological function is typically determined by their 3D structures. However, due to the intrinsic structural heterogeneity caused by the flexible backbones and weak long-range tertiary interactions, it is more challenging to experimentally solve the structure of an RNA than a protein[1]. For example, only ~6000 RNA structures are deposited in the Protein Data Bank (PDB)[2], which is much less than the number of deposited protein structures (~190,000). Thus, there is a great demand for developing efficient algorithms to predict RNA 3D structures.

The current RNA 3D structure prediction methods can be divided into two groups: template-based methods and de novo methods. Template-based methods predict the target structure using homologous templates in PDB. For example, representative methods, such as ModeRNA[3] and MMB[4], work by reducing the sampling space with homologous structures. In general, the predicted structure models by template-based methods are accurate when homologous templates exist in PDB. However, the progress for template-based methods is slow, due to the limited number of known RNA structures and the difficulty of aligning RNA sequences.

On the contrary, de novo methods build 3D conformations by simulating the folding process from scratch. With molecular dynamic simulations and/or fragment assembly, methods such as FARNA[5], FARFAR[6], FARFAR2[7], SimRNA[8], iFoldRNA[9], RNAComposer[10], and 3dRNA[11,12], work well for certain small RNAs (<100 nucleotides). Nevertheless, it is hard to generate accurate 3D structures for large RNAs with

---

[1]School of Mathematical Sciences, Nankai University, Tianjin 300071, China. [2]MOE Frontiers Science Center for Nonlinear Expectations, Research Center for Mathematics and Interdisciplinary Sciences, Shandong University, Qingdao 266237, China. [3]School of Science, Ningxia Medical University, Yinchuan 750004, China. [4]School of Medical Technology, Beijing Institute of Technology, Beijing 100081, China. [5]These authors contributed equally: Wenkai Wang, Chenjie Feng, Renmin Han. ✉e-mail: zhangfa@ict.ac.cn; zhenling@email.sdu.edu.cn; yangjy@sdu.edu.cn

complicated topologies, due to the inaccurate force field parameters and the huge sampling space. To partly address this issue, inter-nucleotide contacts predicted by direct coupling analysis (DCA) have been used to guide the structure simulations[13–15]. In addition, given the hierarchical nature of RNA structure folding, a few methods derive 3D structures from secondary structures, such as Vfold[16,17] and MC-Fold[18]. They are very fast but the modeling accuracy largely depends on the quality of the input secondary structures. The RNA-Puzzles experiments indicate that it remains a grand challenge to accurately predict the structures for large RNAs with complex architectures[19,20].

Deep learning has recently been used to improve de novo RNA 3D structure prediction. The predicted inter-nucleotide contacts by the residual convolutional network (ResNet) are about two times more accurate than DCA, improving 3D structure prediction to some extent[21,22]. It was shown that with the model selection from a geometric deep learning-based scoring system (ARES), the FARFAR2 protocol predicted the most accurate models for four targets in the blind test of the RNA-Puzzles experiments[23]. Recently, inspired by the success of AlphaFold2[24], a few new deep learning-based methods are developed, such as DeepFoldRNA[25], RoseTTAFoldNA[26], and RhoFold[27].

In this work, we introduce trRosettaRNA, an automated deep learning-based approach to RNA 3D structure prediction. It is partly inspired by the successful application of deep learning in protein structure prediction, especially in AlphaFold2[24] and our previous method trRosetta[28–30]. Benchmark tests and blind tests show that trRosettaRNA is promising to enhance RNA structure prediction. The server and source codes are available at: https://yanglab.qd.sdu.edu.cn/trRosettaRNA.

## Results

### Overview of trRosettaRNA

The architecture of trRosettaRNA is depicted in Fig. 1a. Starting from the nucleotide sequence of an RNA of interest, a multiple sequence alignment (MSA) and a secondary structure are first generated by the programs rMSA[31] and SPOT-RNA[32], respectively. They are then converted into an MSA representation and a pair representation, which are fed into a transformer network (named RNA-former, see Fig. 1b and Methods for more details) to predict 1D and 2D geometries (see Fig. S1). Similar to trRosetta, these geometries are converted into restraints to guide the final step of 3D structure folding based on energy minimization (see Methods). Unless otherwise specified, the RMSDs mentioned below are calculated by considering all atoms using the evaluation toolkit provided by the RNA-Puzzles community[33].

### Performance of trRosettaRNA on 30 independent RNAs

To evaluate trRosettaRNA, we collected 30 non-redundant RNA structures based on both release date and similarity with the training RNAs. These RNAs are released after the training RNAs date (i.e., 2017-01) and do not share sequence similarity with trRosettaRNA and SPOT-RNA's training RNAs (see Methods).

We compare trRosettaRNA with two representative methods, RNAComposer[10] and SimRNA[8]. The same secondary structures (from SPOT-RNA) were fed into both methods for fair comparison. For RNAComposer, we submitted the RNA sequences and the predicted secondary structures to its web server to generate 3D models. SimRNA was installed and run locally in our computer cluster. We evaluate the first predicted model to ensure fairness. As shown in Table 1, on these 30 RNAs, the average RMSD by trRosettaRNA (8.5 Å) is significantly lower than those by RNAComposer (17.4 Å; P-value = 1.3E-6) and SimRNA (17.1 Å; P-value = 1.1E-7; the P-values presented in this manuscript were calculated by two-tailed Student's t-tests). trRosettaRNA outperforms RNAComposer and SimRNA for 86.7% and 96.7% of the 30 cases, respectively (Fig. S2a). 20% of the models predicted by trRosettaRNA are with RMSD < 4 Å, whereas no models from RNA-Composer and SimRNA can achieve this accuracy. These data

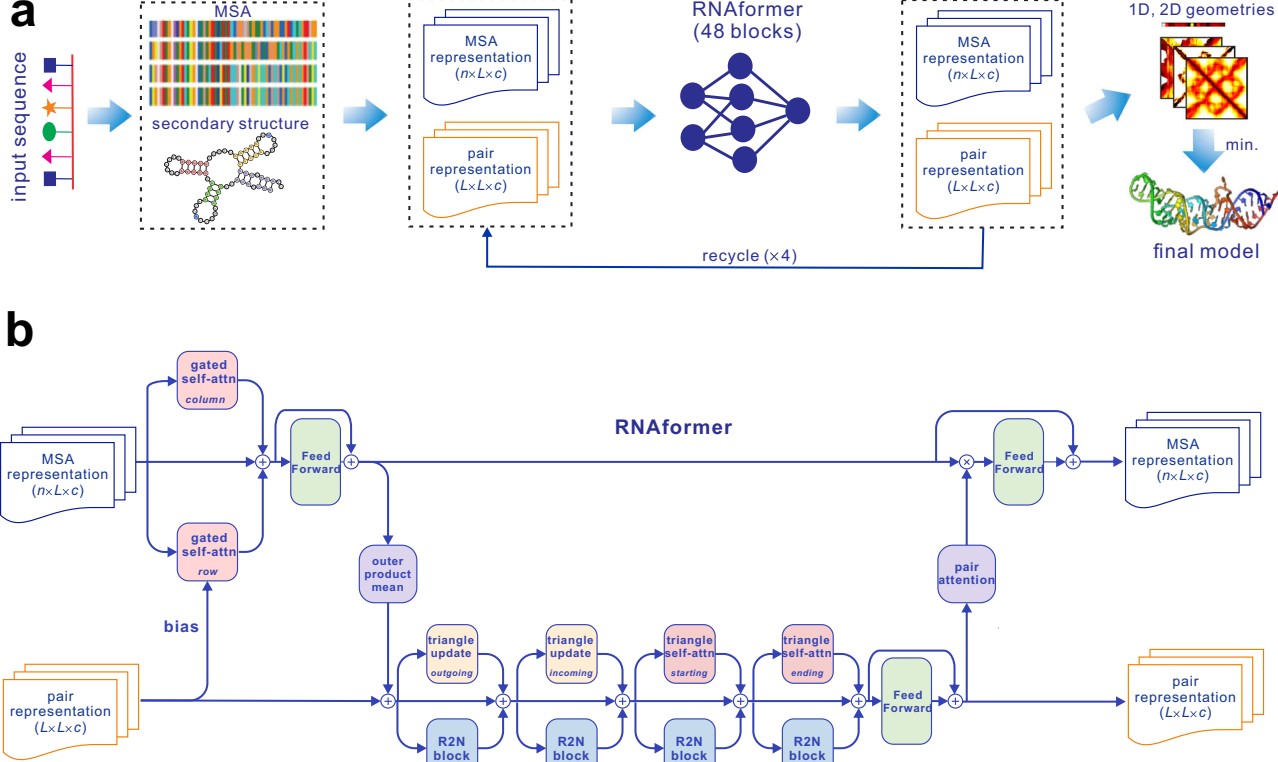

**Fig. 1 | Overall architecture of trRosettaRNA. a** flowchart of trRosettaRNA. **b** structure of each RNAformer block. *n*, *L*, and *c* are the number of sequences in the MSA, the length of the query sequence, and the number of channels, respectively.

**Table 1 | Comparison between trRosettaRNA, SimRNA, and RNAComposer on 30 independent RNAs**

| Methods | Average RMSD (±std.) (Å) | Ratio of accurate models (RMSD < 4 Å) | *P*-value |
|---|---|---|---|
| SimRNA | 17.1 (± 5.2) | 0% | 1.1E-7 |
| RNAComposer | 17.4 (± 6.8) | 0% | 1.3E-6 |
| trRosettaRNA | 8.5 (± 5.7) | 20% | - |

Please note that the evaluation here is on the first model predicted by each method. The *P*-values presented here were calculated by two-tailed Student's *t*-tests. No adjustments were made for multiple comparisons. Source data are provided as a Source Data file.

demonstrate the superiority of the proposed pipeline over traditional RNA structure prediction methods.

We further analyze the impact of the input features (i.e., MSA and secondary structure). The MSA quality is measured by the alignment depth, i.e., the logarithm of the effective number (denoted by $\log(N_{eff})$) of homologous sequences with <80% sequence identity. As shown in Fig. S2b, the RMSD of the trRosettaRNA model is correlated with $\log(N_{eff})$ (Pearson correlation coefficient, PCC = −0.32). trRosettaRNA outperforms RNAComposer and SimRNA at all $\log(N_{eff})$ levels, especially on targets with high $\log(N_{eff})$ values. For the models by RNA-Composer and SimRNA, the correlations between the RMSDs and the alignment depth are weak (PCCs are −0.05 and 0.21, respectively), probably because they do not use MSA during modeling. In contrast, there is a stronger correlation between the RMSDs of the predicted models and the accuracy of the predicted secondary structures (measured by F1-score) for all methods (PCCs are −0.35, −0.31, and −0.29 for trRosettaRNA, SimRNA, and RNAComposer, respectively, Fig. S2c). This is consistent with the observations that precise RNA secondary structure prediction plays a key role in successful 3D structure modeling[19,20].

To provide a more direct demonstration of the contribution of MSA to the RNA structure prediction, we also evaluate the performance of trRosettaRNA when MSAs are excluded. As shown in Fig. S3, for 21 out of the 30 RNAs, the introduction of MSAs helps improve the accuracy of the predicted models. Fig. S4 presents the MSAs for two example RNAs (PDB IDs: 5KH8/7D7V). The coevolutionary information extracted from these MSAs not only covers the majority of 2D base-pairing interactions but also encompasses some 3D interactions (highlighted by green circles in the 2D maps of Fig. S4). With the assistance of MSAs, trRosettaRNA can generate more accurate models for these two RNAs (refer to the bottom right of each subfigure in Fig. S4).

As the secondary structure used in trRosettaRNA is predicted by SPOT-RNA, it is important to consider the possibility of inaccurate predictions by SPOT-RNA. Among the 30 RNAs in the dataset, there are 8 RNAs for which SPOT-RNA failed to predict accurate secondary structures (i.e., F1-score <0.5). Table S1 reveals that for 6 of these 8 RNAs, the secondary structures of trRosettaRNA models are more accurate than those predicted by SPOT-RNA. In Fig. S5a, it is evident that trRosettaRNA corrects certain false positive base pairs, which are highlighted by red circles. Furthermore, trRosettaRNA identifies some interactions that were missed by SPOT-RNA, as indicated by the green circles in Fig. S5a. These observations suggest that trRosettaRNA can correct incomplete or inaccurate secondary structures, although its accuracy is correlated with the quality of the input secondary structures.

Nevertheless, when considering the entire set of 30 RNAs, trRosettaRNA exhibits a slight drop in the average F1-score of secondary structures, decreasing it from 0.65 to 0.6 (as shown in Fig. S5b). This decrease is mainly from the cases where the secondary structures predicted by SPOT-RNA are accurate (F1-score > 0.6). This may be caused by the potential conflicts between the predicted distance

restraints and the base pair restraints, which are not trivial to resolve, especially for targets modeled with low confidence.

However, as a data-driven method, the performance of trRosettaRNA is influenced by the structural homology existing between the target RNA and previously solved RNA structures, which is measured by the maximum TM-score$_{RNA}$. As shown in Fig. S2d, the correlation between the structural homology and the RMSD of trRosettaRNA models is stronger compared to SimRNA and RNA-Composer (PCCs are −0.6, −0.0003, and −0.05, respectively). For five RNAs lacking homolog match (i.e., maximum TM-score$_{RNA}$ with solved RNAs below 0.45), the average RMSD of trRosettaRNA models is 15.8 Å. This value significantly drops to 7.0 Å for the remaining 25 RNAs possessing structural homologs. This discrepancy may be due to the current limitation in the number of solved RNA structures within the PDB database, which in turn impacts the performance of data-driven methods on RNAs with novel structures. Nevertheless, for the 5 RNAs without structural homologs, the average RMSD of the models by trRosettaRNA (15.8 Å) remains lower than SimRNA (20.6 Å) and RNA-Composer (24.3 Å), illustrating the superiority of the deep-learning method over traditional methods for automated prediction.

### Performance of trRosettaRNA on RNA-Puzzles targets

We further test trRosettaRNA on 20 targets from the RNA-Puzzles experiments[19,20]. The target information and the prediction results are summarized in Tables S2 and S3, respectively. It turns out that these targets are harder to predict than the 30 independent RNAs, as revealed by the increased value of RMSD (from 8.5 Å to 10.5 Å).

We compare the trRosettaRNA predictions with the original submissions from the RNA-Puzzles experiments. According to the official assessments[19,20], the most accurate approach is the Das group, which submitted models for 17 targets. Table S3 summarizes the results on these targets for the models from the Das group (denoted by Das) and the best of the models from all groups (denoted by PZ_best). On these 17 targets, the average RMSD of the first predicted models by our method is 10.3 Å, compared with 9.3 Å and 7.2 Å from Das and PZ_best, respectively. For 9 of the 17 targets, the trRosettaRNA models are more accurate than the Das models. Similar observations can be obtained when all the five submitted models are assessed (Table S3). Note that certain participating groups may utilize human experts and/or literature data to guide the modeling during the prediction seasons of the RNA-Puzzles experiments. In contrast, the trRosettaRNA predictions, which are fully automated, achieve similar accuracy to the top human groups.

To obtain a more comprehensive understanding of the quality of trRosettaRNA models, we employ other evaluation measures used in the RNA-Puzzles assessment, in addition to RMSD (Table S4). These measures include the deformation index (DI; evaluating the predicted structures with both RMSDs and base interactions; lower is better)[34], the Interaction Network Fidelity (INF; evaluating the interactions in the predicted structure; higher is better)[34], and MolProbity clash scores (number of serious steric overlaps per 1000 atoms in a structure; lower is better)[35]. The average clash score of the trRosettaRNA models (3.2) is significantly lower than that of the Das models (10.8), indicating that trRosettaRNA not only achieves higher accuracy but also produces higher-quality models. Nevertheless, trRosettaRNA models exhibit worse INF and DI scores than the Das models. This can be explained by the inherent features of our methodology. First, the secondary structures used by trRosettaRNA are predicted rather than from experiments, literature, or human annotation. These predicted secondary structures may provide inaccurate interaction information to the transformer network. Second, trRosettaRNA uses predicted geometry restraints rather than fragment assembly to derive the secondary structures. Both factors may lead to less accurate local base-base interactions in the trRosettaRNA models.

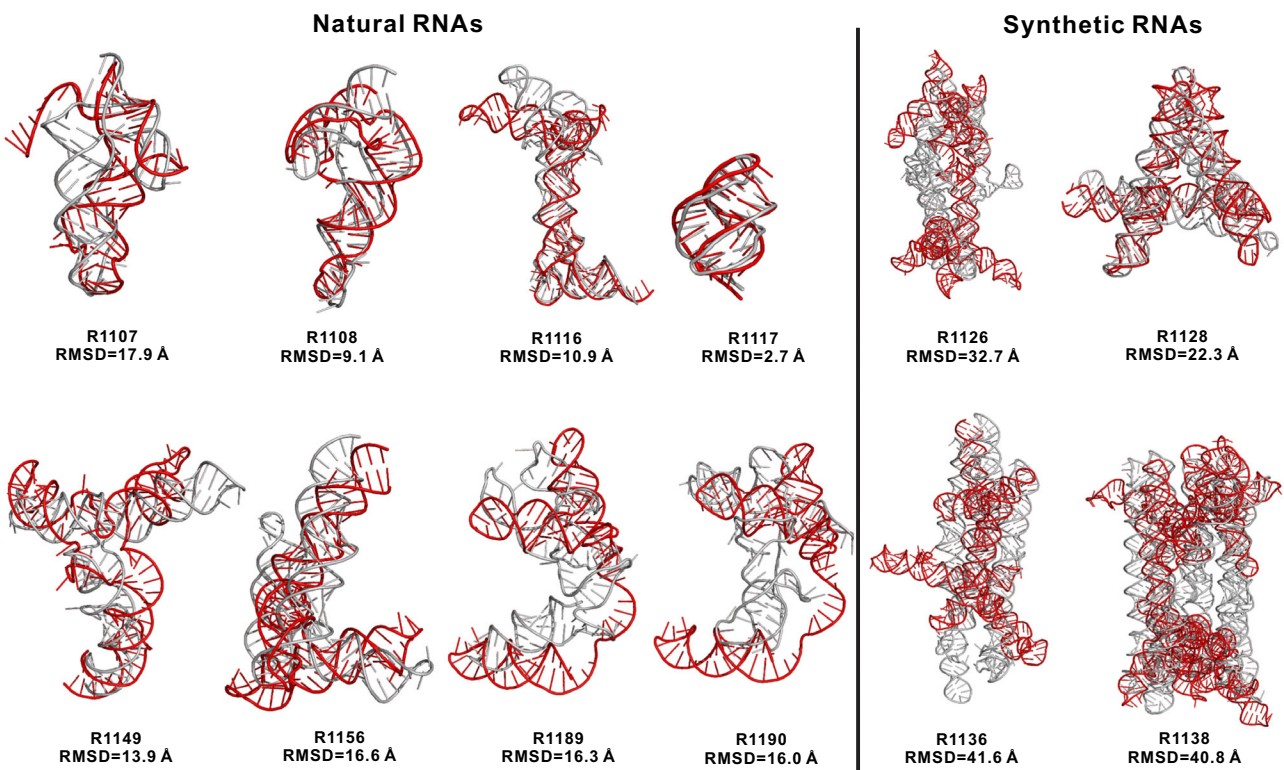

**Fig. 2 | Yang-Server models (red) versus experimental structures (gray) for 12 CASP15 targets.** Consistent with Table 2, the best-submitted models are shown here. The 3D structures are presented using PyMOL.

## Blind test in CASP15

Based on trRosettaRNA, we participated in the blind test of the CASP15 experiment on RNA structure prediction as an automated server (group name Yang-Server). The Yang-Server models for the 12 CASP15 RNAs are shown in Fig. 2. According to the official ranking, Yang-Server is ranked the 9th out of 42 RNA structure prediction groups (including 33 human groups and 9 server groups). Yang-Server is ranked second (after the UltraFold_Server) when considering automated server groups only. Note that the official ranking considers both global and local accuracy, including TM-score$_{RNA}$, GDT-TS score, INF, lDDT, and steric clash; while the main objective optimized in trRosettaRNA is RMSD. Based on the cumulative Z-score of RMSD ( > 0.0), Yang-Server's ranking is improved: 5th/42 for all groups and 1st/9 for server groups (Fig. S6). Yang-Server also achieves a higher ranking than other deep learning-based groups such as AIchemy_RNA (based on RhoFold[27]), BAKER (based on RoseTTAFoldNA[26]), and DF_RNA (based on DeepFoldRNA[25]) in terms of the Z-score of RMSD. According to the RNA-Puzzles assessment[36], the Yang-Server predictions (though not perfect) for two protein-binding targets (R1189 and R1190) are the most accurate among all submitted models (with RMSDs of 16.3 Å and 16.0 Å, respectively). This result demonstrates the potential of our method in predicting protein-binding RNAs even in the absence of binding partner information, though the accuracy is far from satisfactory.

The 12 RNAs in CASP15 can be classified into two categories based on their sources: eight of them are natural, while the remaining four are synthetic. On the eight natural RNAs, Yang-Server yields comparable results to the top human group, AIchemy_RNA2 (mean RMSDs of the first/best in five models: 14.8/12.9 Å versus 15.7/11.3 Å; Table 2 and S5). It is worth noting that trRosettaRNA does not consider structural templates, which may be crucial in improving the modeling accuracy. For example, for the targets, R1107, R1108, and R1149, secondary structure templates can be easily found in the RFAM database (version 14.4, released in December 2020) using an automated process[37]. With these secondary structure templates, trRosettaRNA predicts much more accurate 3D structures than the models submitted during the CASP15 season (Fig. 3). The RMSD values are reduced from 17.9 Å, 9.1 Å, and 13.9 Å to 4.3 Å, 4.8 Å, and 10.6 Å, for R1107, R1108, and R1149, respectively, competitive to the models by AIchemy_RNA2 (i.e., 4.5 Å, 4.5 Å, and 10.5 Å). Thus we believe that the fusion of high-quality secondary structure templates and deep-learning techniques can improve the performance further.

Nevertheless, when it comes to the modeling of synthetic RNAs, there is a notable margin between all the deep learning-based groups (including ours) and the top human groups such as AIchemy_RNA2 (the bottom half of Tables 2 and S5). Note that the top groups for these targets are all based on human-intervened simulations rather than automated modeling. For example, the leading group AIchemy_RNA2 model predicted RNA structure based on the assembly of manually-detected RNA structural motifs followed by full atom optimization with the BRiQ statistical potential[38,39].

The challenge in the automated structure prediction of synthetic RNAs may be explained by a few factors. First, the deep learning-based approach may be biased towards the limited training data, which are mainly from natural RNAs. The synthetic RNAs lack globally homologous RNA sequences and similar structures to the existing RNAs (the maximum TM-score$_{RNA}$ is around 0.3), which may hinder the neural networks from inferring meaning predictions. Second, the human groups were given a three-week deadline for each target, allowing the elaborate human-expert interventions in the modeling procedure. In contrast, the Yang-Server predictions for each target were generated automatically in three days. As a fair comparison, we run the SimRNA package and RNAComposer server with the same secondary structures used by Yang-Server as inputs. The results show no superiority to the Yang-Server models (Fig. S7). This highlights the inherent challenge in automated modeling for these synthetic RNAs, which applies to both conventional and deep learning-based methods.

For example, R1138 contains a few helix hinges and kissing loops, which play important roles in the folding of the overall structure. While

**Table 2 | Results for 12 RNA targets in CASP15**

| Target type | Target ID | RMSD (Å) | | | | | |
|---|---|---|---|---|---|---|---|
| | | Yang-Server | AIchemy_RNA2 | Chen | RNApolis | Deep learning best[a] | Overall best |
| Natural | R1107 | 17.9 (4.3[b]) | 4.5 | 6.5 | 8.8 | 5.9 | 4.5 |
| | R1108 | 9.1 (4.8[b]) | 4.5 | 6.0 | 8.5 | 4.8 | 4.5 |
| | R1116 | 10.9 | 17.3 | 18.0 | 12.7 | 7.9 | 4.8 |
| | R1117 | 2.7 | 2.3 | 2.0 | 2.7 | 2.7 | 2.0 |
| | R1149 | 13.9 (10.6[b]) | 10.5 | 14.0 | 18.2 | 6.9 | 6.9 |
| | R1156 | 16.6 | 7.6 | 11.0 | 17.1 | 12.9 | 5.4 |
| | R1189 | 16.3 | 22.0 | 21.2 | 18.7 | 22.8 | 16.3 |
| | R1190 | 16.0 | 22.0 | 18.8 | 22.4 | 22.2 | 16.0 |
| | Average | 12.9 (10.3[b]) | 11.3 | 12.2 | 13.6 | 10.8 | 7.5 |
| Synthetic | R1126 | 32.7 | 8.8 | 12.6 | 20.0 | 30.2 | 8.9 |
| | R1128 | 22.3 | 4.3 | 6.7 | 14.6 | 14.3 | 4.3 |
| | R1136 | 41.6 | 7.3 | 10.9 | 11.0 | 27.3 | 7.2 |
| | R1138 | 40.8 | 7.8 | 12.3 | 9.6 | 35.5 | 7.8 |
| | Average | 34.4 | 7.0 | 10.6 | 13.8 | 26.8 | 7.0 |
| Overall average | | 20.1 (18.3[b]) | 9.9 | 11.2 | 13.7 | 16.1 | 7.4 |

[a]According to the CASP15 abstracts, there are 14 RNA prediction groups utilizing deep learning-based methods to predict RNA structures.
[b]trRosettaRNA results with secondary structure templates as inputs.
For all compared groups, we evaluate their best-submitted models for each target. The evaluation based on the first predicted model is shown in Table S5.

the automated methods successfully establish satisfactory local interaction networks (with INF value of ~0.7), the predicted models still deviate significantly from the experimental structure when considering the global 3D topologies (Fig. S8b). Accurately predicting the kissing loops (such as the one highlighted by the black circle) and the helix hinges in R1138 poses a significant challenge for automated methods, despite the recurrence of these motifs across numerous RNA structures. To illustrate, trRosettaRNA correctly predicted the highlighted kissing loop (highlighted by the black circle on the predicted distance map in Fig. S9b). Utilizing this limited set of distance restraints, trRosettaRNA can successfully generate an accurate structure of the kissing loop (Fig. S9a). However, this particular kissing loop was not predicted correctly when modeling the structure globally, probably due to the complicated interactions between other motifs (Fig. S9c). This reflects the modeling difficulty of such synthetic RNA by automated approaches. As mentioned by AIchemy_RNA2, the accurate modeling of these synthetic RNAs requires extensive human-expert interventions, involving template detection, secondary structure determination, motif assignment, and more[39].

As the primary focus of our method is to optimize the global RMSD, it is also worthwhile to investigate other metrics (Table S6). In addition to the INF and MolProbity clash score mentioned above, we also evaluate the Local Distance Difference Test (lDDT[40]) score, which measures the local inter-residue distance error and has been widely used to evaluate protein structure models. In terms of the two local metrics (INF and lDDT), the Yang-Server models show a notable margin with AIchemy_RNA2 for both natural and synthetic RNAs. This discrepancy can be attributed to the reasons mentioned earlier, namely, the inaccurate input secondary structures and the methodological differences between the two approaches (deep learning-based versus fragment assembly-based). The integration of local fragment segments and deep-learning techniques is promising for bridging this gap in the future.

During CASP15, our method did not consider steric clashes in modeling, resulting in high clash scores (>20) for our submitted models. This might impact our official ranking which considers both modeling accuracy and steric clashes. According to the CASP15 assessments[36], Yang-Server is ranked 26th out of the 42 groups in terms of clash score, worse than AIchemy_RNA and BAKER. We addressed this issue after CASP15 by implementing an additional refining step at the end of the energy minimization procedure. Consequently, the average clash score of the 12 CASP15 targets dropped significantly from 33.84 to 3.05, which is much lower than that of AIchemy_RNA2 (16.68).

## Blind test on the latest RNA-Puzzles targets
In addition to our participation in CASP15, we also took part in the blind tests of three RNA-Puzzles targets as an automated server group named Yang. These targets include PZ37 (PDB ID: 8GXC; a ligand-binding dimer), PZ38 (PDB ID: 8HB8; a ligand-binding riboswitch), and PZ39 (PDB ID: 8DP3; a protein-binding cloverleaf RNA). The results are summarized in Fig. 4a, b.

In the case of PZ37/PZ38, our results (RMSD 10.3 Å/8.7 Å) are highly competitive, ranking at 3/3 out of 16/15 participating groups, and only surpassed by the human groups Chen and Szachniuk. It is worth noting that our predictions were fully automated and did not consider the ligand or dimer information, making our results impressive. However, we notice that the first model chosen for PZ38 is the worst among the five submitted models, with an RMSD of 14.4 Å, compared to 8.7 Å of the best model. This reflects that while trRosettaRNA exhibits the capability to produce models with commendable accuracy, there remains potential for model ranking.

For the target PZ39, the RMSD of our models are higher than 15 Å. PZ39 has no similar sequence (according to BLASTN search at an e-value cutoff of 10) nor similar structure (according to TM-score$_{RNA}$ > 0.45) from the known RNAs. This may account for the poor performance of our method on this target. Nonetheless, local motif templates can be found for this RNA. For example, for the fragment consisting of residues 20 to 26 which forms the Fab binding site, it is easy to identify the same fragments in known Fab-binding RNAs (e.g., in PDB IDs: 6DB9 and 3IVK). Given the limited data of available RNA 3D structure, a promising approach to improving may be based on the combination of deep learning with conventional physics-based and/or fragment assembly-based methods.

## Comparison with other deep learning-based methods
During the preparation of this manuscript, another three deep learning-based approaches (DeepFoldRNA, RoseTTAFoldNA, and RhoFold) were posted. As mentioned above, trRosettaRNA achieves a higher summed Z-score of RMSD than these methods in the blind test

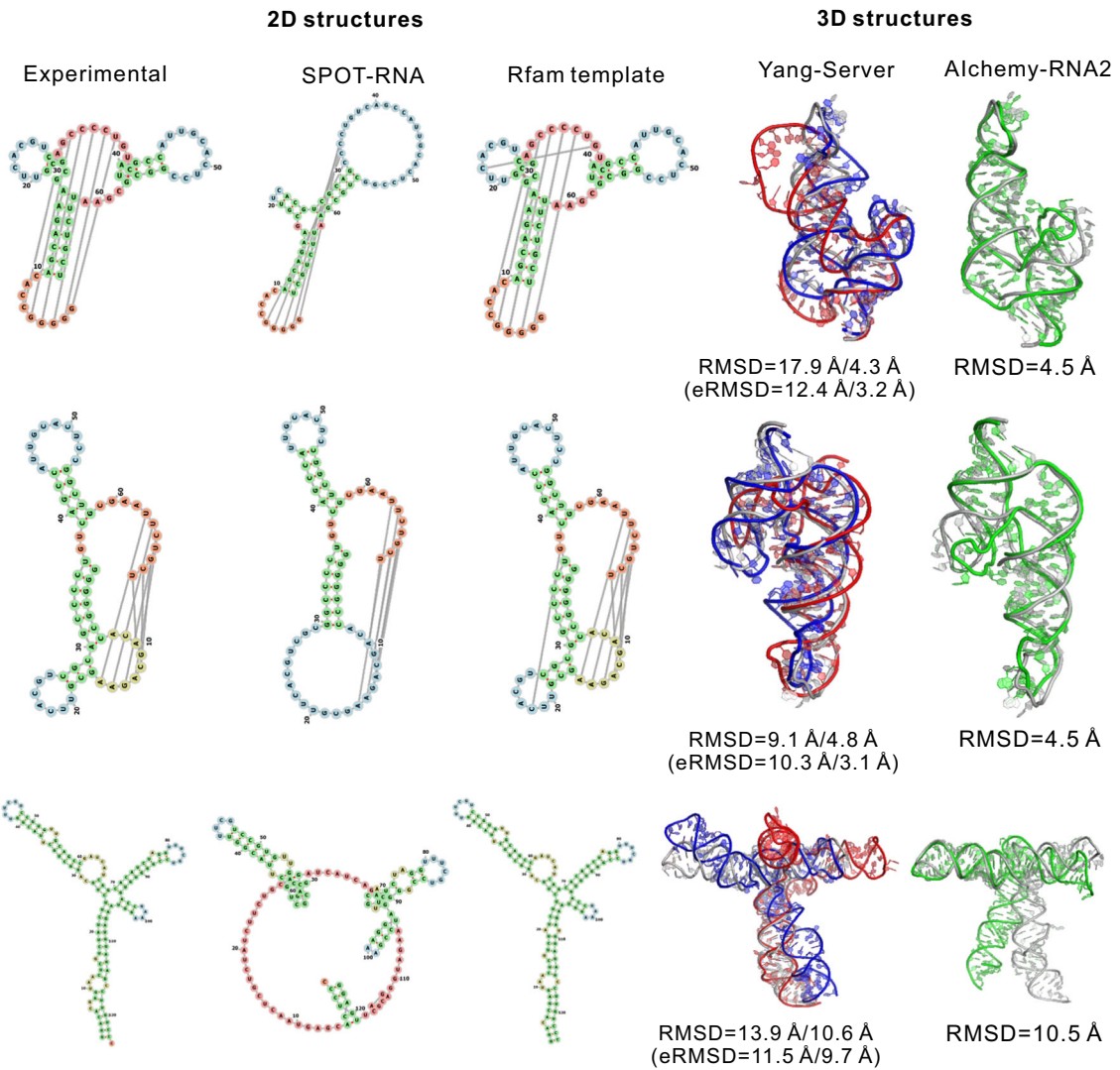

**Fig. 3 | Results for three targets from CASP15 for which the template secondary structures can be found in the Rfam database.** The RNA secondary structure visualization was employed with forna[52]. The template search and 2D structure modelling were employed with R2DT program[37]. For the 3D modelling results, we present the best model submitted by Yang-Server (in red), the trRosettaRNA model based on 2D templates (in blue) and the Alchemy_RNA2 best model (in green). Both predicted 3D structures are superimposed onto the experimental structures (gray). For Yang-Server models, the RMSD and eRMSD values are shown in SPOT-RNA-based/R2DT-based format.

of the CASP15 competition. We further conducted head-to-head comparisons between these methods on RNAs from the blind tests (CASP15 and RNA-Puzzles). For each target from blind tests, we used the result of the first submitted models if available; otherwise, we ran the program locally to predict the structure model.

The results show that trRosettaRNA achieves a 3.3/2.1 Å lower RMSD than DeepFoldRNA/RoseTTAFoldNA (orange/purple points in Fig. 4c) on the 15 RNAs from the blind tests. For 11/9 out of these 15 RNAs, the trRosettaRNA predictions are more accurate than those by DeepFoldRNA/RoseTTAFoldNA. The average RMSD of trRosettaRNA models (21.3 Å) is marginally higher than RhoFold models (20.6 Å; P-value = 0.9; blue points in Fig. 4c), which is inconsistent with the comparison based on the Z-score of RMSD (i.e., Fig. S6). This slight difference is mainly due to the poor performance of trRosettaRNA on two CASP15 RNAs (17.9 Å for R1107 and 9.1 Å for R1108; compared to 5.9 Å and 5.4 Å for RhoFold, respectively; highlighted by red circle in Fig. 4c). As mentioned above (see also Fig. 3), this performance gap can be effectively bridged by employing more confident secondary structures as inputs. For the remaining 13 RNAs, the average RMSD of trRosettaRNA (22.5 Å) is slightly lower than RhoFold (22.9 Å).

Moreover, trRosettaRNA outperforms RhoFold for 8 out of the remaining 13 RNAs.

To summarize, trRosettaRNA outperforms DeepFoldRNA and RoseTTAFoldNA, and is competitive with RhoFold in blind tests, highlighting its robustness.

**Impact of the predicted 1D and 2D geometries**

The geometries predicted by the RNAformer network consist of 1D orientations and 2D contacts, distances, and orientations (Fig. S1). To analyze their contributions, we compare the modeling results using different geometries on the 30 RNA-Puzzles targets (Fig. 5a and Table S7). Using the 2D distance restraints only, trRosettaRNA achieves a reasonable RMSD of 11.34 Å. This value is reduced to 10.79 Å when the 1D and 2D orientations are included. Furthermore, with the help of 2D contacts, the RMSD drops to 10.51 Å. We use the target PZ11 (PDB ID: 5LYS) to show the impacts of different restraints. As shown in Fig. 5b, using 2D distance restraints only, trRosettaRNA can generate a structure model with an RMSD of 10.5 Å. However, the helix at the 5'-end and 3'-end (highlighted by the green square in Fig. 5b) is wrongly twisted. The introduction of 1D and 2D orientations fixes the wrong twist of this

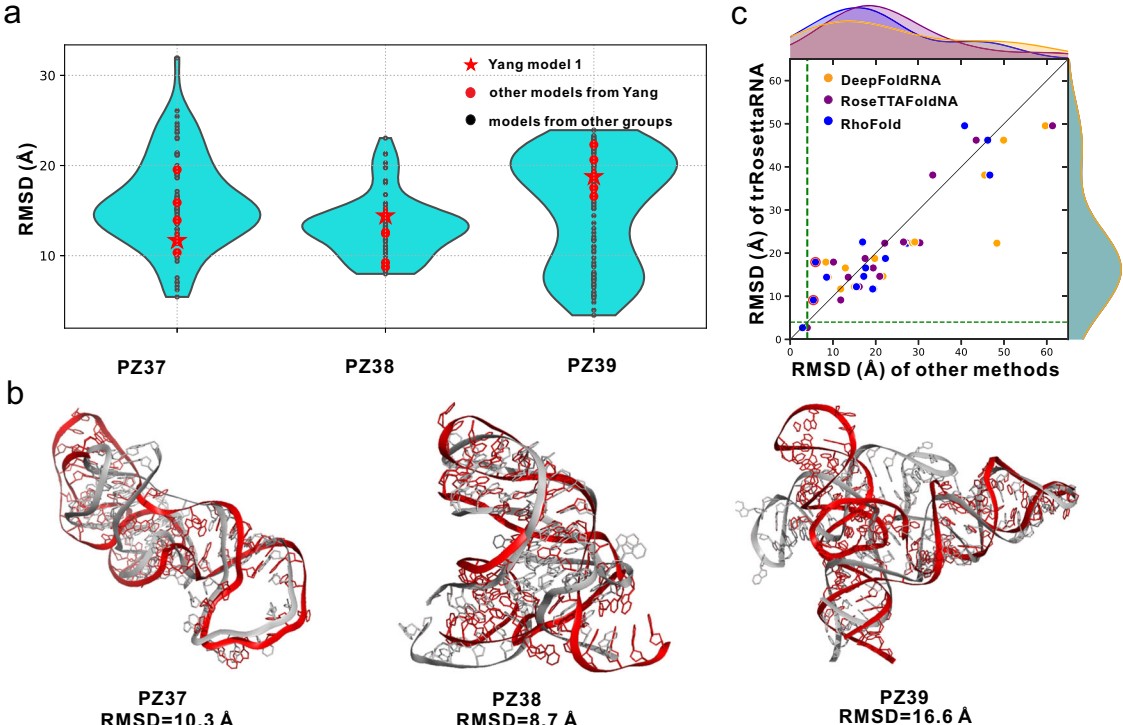

**Fig. 4 | Blind test results and comparison with other deep learning-based methods. a, b** blind test results on the latest three targets from RNA-Puzzles. **a** RMSD comparison of the models submitted by Yang group and models from other groups. **b** the best models submitted by Yang group (red) superposed to the experimental structures (gray). **c** head-to-head RMSD comparison between trRosettaRNA and other deep learning-based methods ($n = 15$ RNAs from the blind tests of CASP15 and RNA-Puzzles). The dashed horizontal and vertical lines correspond to an RMSD of 4 Å. The bar plots show the RMSD distributions. The red circles highlight the two cases (R1107 and R1108) in which trRosettaRNA can achieve better results with improved secondary structures. Source data are provided as a Source Data file.

region. The 2D contact restraints further refine the structure, resulting in a more accurate model with 6.7 Å RMSD.

## Confidence score of the predicted structure models

To guide real-world application, the confidence scores of the predicted protein structure models have been estimated reliably in trRosetta[28–30]. A similar estimation can be extended to trRosettaRNA. Specifically, we first calculate a few variables reflecting the confidence of the predicted distance maps and the convergence of the first structure models (see Methods for more details). Then a linear regression on these variables is employed to fit the RMSD values. For the RNAs from the benchmark datasets, the estimated RMSDs (eRMSDs) correlate well with the real RMSDs of the predicted models (PCC = 0.56, Fig. 5c). Moreover, the eRMSD metric also roughly reflects the modeling difficulty for the 12 CASP15 targets, with an average value of 17.2 Å.

As a practical application, for the three CASP15 RNAs (R1107, R1108, and R1149) with reliable secondary structure templates, the defined eRMSD effectively captures the improvements from the introduction of these templates (Fig. 3 and Table S5). Additionally, in 6 out of the 8 cases where SPOT-RNA provided inaccurate secondary structures, the eRMSD successfully helps identify models with more accurate input of secondary structures (Table S1). These observations highlight the promising potential of eRMSD in facilitating the optimal selection between predictions from various inputs.

## Analysis of the running time

We decompose the running time of trRosettaRNA into two parts: 2D geometry prediction and 3D structure generation. The time for MSA generation is not discussed here as it can be flexible depending on the searching algorithms and sequence databases. Fig. 5d shows that

trRosettaRNA spends most time in the generation of 3D structure (> 95%). With the increase in sequence length, the running time increases linearly. In general, it takes <30 min to complete the prediction for a typical RNA with <200 nucleotides.

## Application to Rfam families with unknown structures

It remains challenging to solve RNA structures by experiment. For example, only 123 out of the 3938 families in the Rfam database (version 14.4) have experimentally resolved 3D structures[41]. We sought to predict the structures for the Rfam families that have no experimental structures. We collected 1752 unsolved families that are 50–200 nucleotides long and have more than 30 members. For each family, we use its consensus secondary structure along with the MSA derived from the consensus sequence as the input features to trRosettaRNA. Most of these families are not predicted well, with eRMSD > 10 Å for 891 of 1752 families (Fig. 6a). This may reflect the difficulty of determining the structures for these families.

Nevertheless, trRosettaRNA does predict accurate structures for 263 families with eRMSD <4 Å. For 27 of these families, the predicted structure models do not have any similar structures in PDB according to the program RNAalign (TM-score$_{RNA}$[42] ≥ 0.45). In Fig. 6b, we show the predicted structures for 6 families with distinct topologies. These high-confidence models are anticipated to provide a structural basis for understanding their biological functions and guide their experimental determinations. For example, for the family sul1 RNA (RF01070) which encodes a subunit of an enzyme participating in the citric acid cycle, trRosettaRNA can generate a confident model with an estimated RMSD of 1.6 Å. The trRosettaRNA models for the 263 families with eRMSD <4 Å are available on our website (https://yanglab.qd.sdu.edu.cn/trRosettaRNA/rfam/).

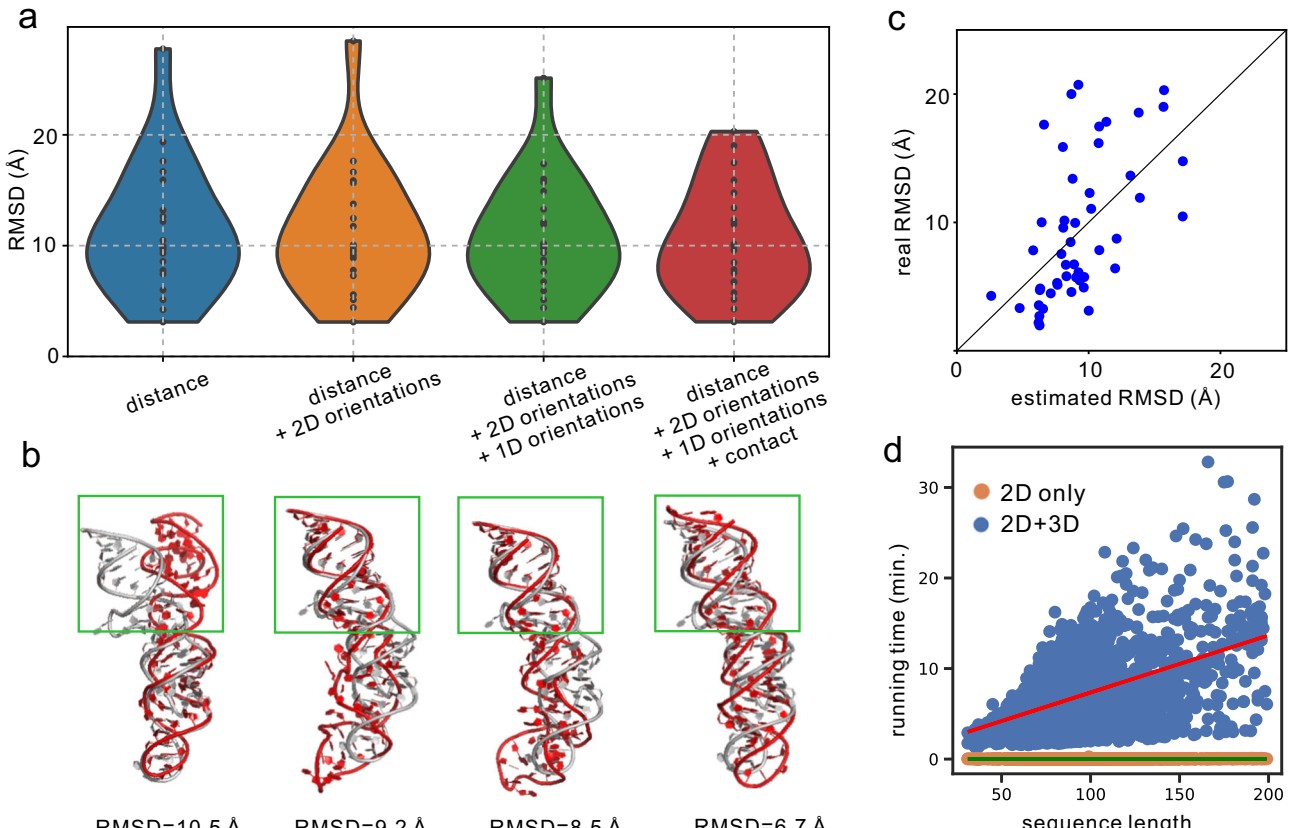

**Fig. 5 | Summary of the folding results by different restraints. a** contribution of the various restraints to the trRosettaRNA modeling accuracy in terms of the RMSD for the 20 RNA-Puzzles targets ($n = 20$ RNAs). **b** an example (PZ11) to illustrate the impact of different restraints. The predicted models (red cartoon) are superposed to the experimental structures (gray cartoon). The green square highlights the helix region which is influenced by the introduction of more restraints. **c** head-to-head comparison between the estimated and real RMSD for all RNAs in the benchmark datasets ($n = 50$ RNAs). **d** the relationship between the running time and the sequence length on 1752 Rfam families. The 2D geometry predictions (orange dots) were run on one GPU card. The 3D structure folding was performed on one CPU core. Source data are provided as a Source Data file.

## Discussion

We have developed trRosettaRNA, an automated approach to RNA 3D structure prediction with the transformer network. We have rigorously assessed trRosettaRNA with two independent datasets and two blind tests. The benchmark tests show that trRosettaRNA predicts more accurate models than the other automated methods. trRosettaRNA was assessed blindly in two experiments: RNA-Puzzles (3 targets) and CASP15 (12 targets). The RNA-Puzzles experiments show that the automated predictions by trRosettaRNA are competitive with the top human predictions for 2 out of 3 targets. The CASP15 experiments show that trRosettaRNA outperforms other deep learning-based methods in terms of the cumulative Z-score based on RMSD. Our method achieves comparable accuracy to the top human groups on 8 natural RNAs, though without any human interventions.

However, we notice that the average RMSD on the natural RNAs from the CASP15 blind test (14.8 Å for the first models) is higher than that on the RNAs from the two benchmark datasets (8.5 Å for 30 independent RNAs and 10.5 Å for 20 previous RNA-Puzzles targets). The disparity in the modeling accuracy may be explained by the target difficulty and novelty. (1) target difficulty. Most of the CASP15 RNAs exhibit high flexibility and can adopt multiple conformations (except for R1116 and R1117)[36]. In addition, there are two dimers (R1107, R1108) and two protein-binding RNAs with many single-strand regions (R1189, R1190). These features pose challenges for SPOT-RNA in predicting confident secondary structures. To illustrate, the average F1-score of the predicted secondary structure by SPOT-RNA is much lower for the

8 natural RNAs from CASP15 in contrast to the 20 RNA-Puzzles targets (0.62 and 0.72, respectively). (2) target novelty. A significant proportion of RNAs (two-thirds, 20 out of 30) from the non-redundant benchmark dataset exhibit high similarities (TM-score$_{RNA} > 0.6$) to previously known RNAs, making them easy to predict for data-driven methods like trRosettaRNA. On the contrary, none of the RNAs from CASP15 show such a level of similarity (Fig. S10).

This reflects the limitations associated with trRosettaRNA and the benchmark tests employed in this work. First, the performance of trRosettaRNA is susceptible to the quality of predicted secondary structures. Secondly, though trRosettaRNA achieves promising accuracy in the internal benchmark tests, its performance on novel RNAs remains limited. Moreover, the automated structure prediction of synthetic RNAs remains challenging.

The blind tests in CASP15 experiments suggest that the deep learning approach to RNA structure prediction is still in its infancy. Nevertheless, with consistent development, deep learning should be promising to advance RNA structure prediction. Incorporation of physics-based modeling into deep learning is one of the directions to improve in the future. One of the most instant alternatives is to combine it with other conventional approaches and optimize the algorithms toward those under-represented RNA structures in the future. For example, to overcome the bias toward known RNA folds, neural networks (such as with physics-informed neural networks[43]) can be utilized to learn force fields or to recognize/assemble local motifs instead of directly predicting the global 3D structures.

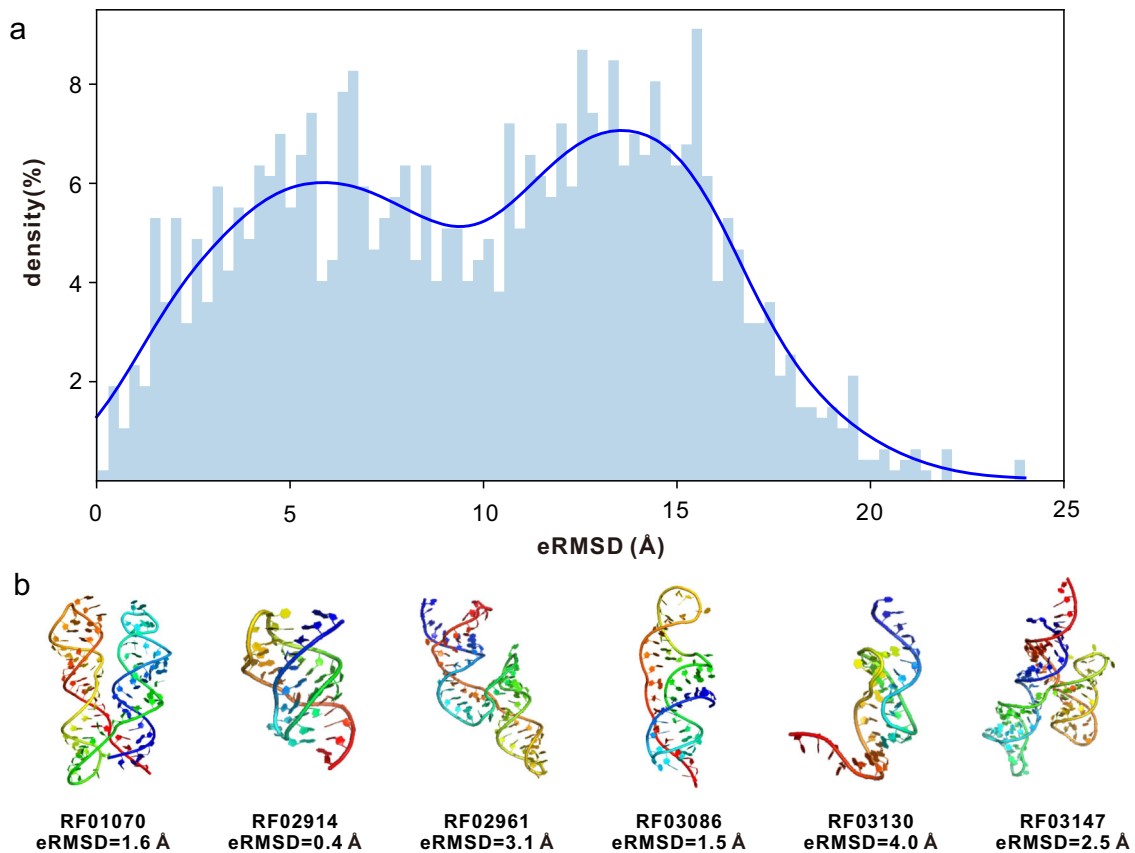

**Fig. 6 | Application of trRosettaRNA to Rfam families with unknown structures. a** eRMSD (i.e., estimated RMSD) distributions of the predicted structure models ($n = 1752$ Rfam families). **b** six selected example families with eRMSD <4 Å. Source data are provided as a Source Data file.

## Methods

### trRosettaRNA algorithm

As shown in Fig. 1a, the full pipeline of trRosettaRNA consists of three major steps: preparation of input data, prediction of 1D and 2D geometries, and generation of 3D structure.

### Step 1. Preparation of input data

For a given query RNA, the first step of trRosettaRNA is to prepare an MSA and a secondary structure. Two different MSAs are generated for each query sequence. The first is generated by using the program rMSA against multiple sequence databases (NCBI's nt, Rfam, and RNAcentral[44]). The second is obtained by running the program Infernal[45] against the smaller database RNAcentral with two iterations, which is very fast. Then we select the final MSA based on the qualities of the predicted distance maps (measured by the average of standard deviations of the probability values of each nucleotide pair, Fig. S11). The secondary structure is predicted by SPOT-RNA[32] from the query sequence. Here we use the predicted probability matrix as the input, which contains more information than the dot-bracket representation.

### Step 2. Prediction of 1D and 2D geometries

The second step of trRosettaRNA is to predict the 1D and 2D geometries by deep learning. We design a transformer network (named RNAformer) similar to the network Evoformer in AlphaFold2. At the very start, the input MSA and secondary structure are converted into two representations, i.e., the MSA representation (i.e., MSA embedded by nucleotide types) and the pair representation (including the direct couplings derived from MSA and the probability matrix of the predicted secondary structure). We adopt a transformer-based module (i.e., RNAformer) to update both representations. More specifically,

each block of RNAformer can be divided into four steps according to the update direction (Fig. 1b).

1. **MSA to MSA**. To update the MSA representation by itself, we perform row- and column-wise gated self-attention operations and combine the corresponding results. A feed-forward layer is employed to introduce nonlinearity. Note that the pair information participates in the row-wise attention by adding bias to the attention maps.

2. **MSA to pair**. We perform an outer product operation on the self-updated MSA representation to transform it into the pair format. In detail, the MSA representation is linearly projected to a smaller dimension. Then for the nucleotide pair $(i, j)$, the outer products of the vectors from the $i^{th}$ and the $j^{th}$ columns of the MSA representation are averaged over the homologous sequences to update the representation for this pair.

3. **Pair to pair**. After the above step, we perform the triangle updates, followed by a feed-forward layer. For each triangle update layer, we use a multi-scale network Res2Net[46] to enhance the ability to model the local details.

4. **Pair to MSA**. The updated pair representation is then linearly projected to the pair-wise attention maps, which are then multiplied on the MSA representation, followed by a feed-forward layer.

A single-pass RNAformer consists of 48 blocks, which are cycled 4 times in the complete inference (Fig. 1a). The final predicted probability distributions of the 2D geometries are derived from the updated pair representation via linear layers and softmax operations. To predict the 1D geometry, we transform the MSA representation into 1D representation by row-wise weighted summation, followed by linear layers and softmax operations to obtain the predicted probabilities.

## Step 3. Generation of full-atom structure models

Similar to trRosetta, trRosettaRNA generates full-atom structure models by energy minimization with deep learning potentials and physics-based energy terms in Rosetta.

$$E = w_1 E_{dist} + w_2 E_{ori} + w_3 E_{cont} + w_4 E_{ros} \quad (1)$$

$$E_{ori} = E_{ori,2D} + \frac{L}{2} E_{ori,1D} \quad (2)$$

where $E_{dist}$, $E_{ori}$, and $E_{cont}$ represent the distance-, orientation-, contact-based restraints and Rosetta's internal energy terms, respectively; $E_{ori,2D}$ and $E_{ori,1D}$ represent the restraints from 2D and 1D orientations, respectively; $L$ is the length of the sequence. A detailed description of these energy terms is available in the Supporting Information. The weights ($w_1 = 1.03$, $w_2 = 1.0$, $w_3 = 1.05$, $w_4 = 0.05$) are decided on hundreds of RNAs randomly selected from the training set to minimize the average RMSD. Note that we only select a subset of restraints with probabilities higher than a specified threshold (0.45, 0.65, and 0.6 for distances, orientations, and contacts, respectively).

The folding procedure is implemented with pyRosetta[47]. From each RNA, 20 full-atom starting structures are first generated using the RNA_HelixAssembler protocol in pyRosetta[47]. The Quasi-Newton-based optimization L-BFGS is then applied to refine these structures by minimizing the total energy, resulting in 20 refined full-atom structure models. Finally, the model with the lowest total energy (Eq. 1) is selected as the final prediction.

## Construction of datasets

**Test sets.** Two benchmark datasets are constructed in this work. The first one is from the RNA-Puzzles experiments. This set consists of all RNA-Puzzles targets from PZ1 through PZ33 except PZ2. PZ2 is a complex that has complicated interactions among eight chains, which is out of the prediction scope of the current work. The second dataset comes from PDB. In detail, we first collected 339 RNA structures from PDB that were released after 2017-01. RNAs with more than 200 or less than 30 nucleotides were removed. Then the program cd-hit-est[48] was used to remove redundant sequences at 80% sequence identity. To avoid over-estimation, RNAs with an e-value lower than 10 by BLASTN searching against the training sets of trRosettaRNA and SPOT-RNA were excluded from both test sets. The duplicated RNAs between these two test sets were also removed. The resulting sets comprised 20 RNA-Puzzles targets and 30 non-redundant RNAs, respectively.

**Training sets from PDB.** To train our models, we first collected all the RNA chains released before 2022-01 in PDB. Multi-chain structures were separated into single-chain structures. Modified nucleotides are replaced by the standard ones. In addition, if two chains form more than three base-pairing interactions, they are linked by three Adenines, resulting in a new sample. In total, we obtained 8849 samples. Then we tried to generate MSA for each query sequence and removed the sequences without sequence homologs. Finally, 3633 RNA chains were retained for training the network models of trRosettaRNA.

To avoid data leakage in the benchmark tests while keeping as many training samples as possible, five training subsets were obtained by filtering the above 3633 RNA chains. Specifically, for the RNA-Puzzles set, we split the 20 RNAs into four subsets according to their release dates in PDB (i.e., 2010-12 ~ 2013-07, 2013-07 ~ 2016-07, 2016-07 ~ 2019-04, and after 2019-04, see Table S2). Correspondingly, four smaller training sets (1133, 1528, 2337, and 3001 samples, respectively) were obtained by removing structures that were released after the above dates. We trained four network models with these training sets, respectively. For each group of the RNA-Puzzles targets, the predictions were made by the model trained on the corresponding training

set. For the 30 independent RNAs, the training set consists of 2454 RNAs that were released before 2017-01.

**Self-distillation training set from bpRNA.** As the number of available RNA structures is limited, inspired by the success of the self-distillation method used in AlphaFold2, we constructed a self-distillation dataset from the bpRNA database with experimental secondary structures[49]. In detail, we collected the bpRNA sequences that are available in the Rfam database[41] so that the Rfam MSAs can be used immediately. Then we removed the orphan families (i.e., with one RNA sequence only) and ran cd-hit-est to exclude the redundant sequences at a sequence identity cutoff of 80%. The final self-distillation dataset consists of 13202 RNA chains. The RNAs possessing an e-value lower than 10 (by BLASTN) or with a sequence identity higher than 80% (by cd-hit-est) with the two benchmark datasets were excluded from the self-distillation dataset when training the models for benchmark tests. Consequently, the self-distillation dataset for benchmark tests consists of 13175 RNA chains.

We use a single un-distilled RNAformer model, i.e., trained on the PDB dataset (or the corresponding subsets for benchmark tests), to generate the predicted labels for the self-distillation set. Using this un-distilled model, we predicted the 1D and 2D geometries (in the form of probability distributions) for every sequence in the self-distillation set. These predicted geometries are then assigned as the labels of these distillation samples. As the predictions may be inaccurate for some nucleotides, we estimate the prediction confidence and filtered out the potentially inaccurate nucleotides and nucleotide pairs. In detail, for each pair of nucleotides ($i$, $j$) with sequence separation less than 128 (i.e., $|i-j| \leq 128$), we computed the mean P-P distance distribution (i.e., the reference distribution, denoted by $P^{ref}_{|i-j|}$), using the predicted distance maps for 1000 samples randomly selected from the self-distillation set. Then for each pair of nucleotides in a self-distillation sequence, we calculated its confidence score (denoted by $c_{i,j}$), defined as the Kullback-Leibler divergence between its predicted distribution (denoted by $P_{i,j}$) and the reference distribution:

$$c_{i,j} = D_{KL}\left(P_{i,j} | P^{ref}_{|i-j|}\right) \quad (3)$$

The per-nucleotide confidence score $c_i$ was calculated as the average of $c_{i,j}$ over all $j$s within the sequence separation of 128:

$$c_i = \frac{1}{128} \sum_{j=i+1}^{i+128} c_{i,j} \quad (4)$$

During training (see below), the nucleotides/nucleotide pairs with confidence scores <0.5 are masked out when calculating the 1D/2D losses, respectively.

## Training procedure and loss function

The training of an RNAformer model can be divided into three steps. In the first step, we trained an un-distilled model using the PDB set by 15 epochs. This model was then used to generate the labels for RNAs in the self-distillation set. In the second step, the un-distilled model was further trained on the combination of the PDB set and the self-distillation set with another 15 epochs. At each epoch, the training samples consist of all the $N$ samples from the PDB set and randomly selected $3N$ samples from the self-distillation set, where $N$ is the size of the PDB set. In the third step, we finetuned the models on the long sequences (>100 nucleotides) selected from the PDB set. We used the Adam optimizer to minimize the loss function (see below) with different learning rates (0.0001 for the first two steps, 0.00005 for the third step).

For all training steps, the loss function is defined as the cross entropy between the predicted distributions and the real or generated

labels. In total, the loss function can be written as:

$$Loss = L_{2D} + L_{1D} + 5L_{cont} \tag{5}$$

where $L_{2D}$, $L_{1D}$, and $L_{cont}$ are the loss items for the 2D distances and orientations, 1D orientations, and 2D contacts, respectively. More specifically, the three loss items can be written as:

$$L_{2D} = \frac{1}{10N_{nt}^2} \sum_{i=1}^{L} \sum_{j=1}^{L} \sum_{g \in \{2D\,geometries\}} CE\left(P_{i,j}^g, Y_{i,j}^g\right) \tag{6}$$

$$L_{1D} = \frac{1}{4N_L} \sum_{i=1}^{L} \sum_{g \in \{1D\,geometries\}} CE\left(P_i^g, Y_i^g\right) \tag{7}$$

$$L_{cont} = \frac{1}{L^2} \sum_{i=1}^{L} \sum_{j=1}^{L} CE\left(P_{i,j}^{cont}, Y_{i,j}^{cont}\right) \tag{8}$$

where $CE()$ is the cross entropy function; $P_{i,j}^g$ is the predicted probability distribution of the 2D geometry $g$ between nucleotides $i$ and $j$; $P_i^g$ is the predicted probability distribution of the 1D geometry $g$ of nucleotide $i$; $P_{i,j}^{cont}$ is the predicted probability of nucleotides $i$ and $j$ to be in contact; the $Y$ heads are the one-hot encodings of the true labels (for PDB samples) or the predicted distributions (for self-distillation samples); $L$ is the number of nucleotides in sequence; 10 and 4 are the number of types of 2D geometries (5 distances + 5 orientations) and 1D geometries (4 orientations), respectively.

### Estimation of model confidence

To estimate the quality of the predicted model, a few variables are first derived from predicted distance maps and generated decoys.

1. *pRMSD*: the average pair-wise RMSD of the top ten decoys with the lowest total energies.
2. *mp*: the mean probability of the predicted inter-nucleotide distances for the set (denoted by $S$) of the top $15L$ ($L$ is the sequence length) nucleotide pairs (as ranked by the probability $P(d_{P\text{-}P} < 40\,\text{Å})$). A similar variable has been defined to estimate the accuracy of predicted inter-residue distances[50].

$$mp = \frac{1}{N_{bins}} \sum_{k=1}^{N_{bins}} \frac{1}{|M_k|} \sum_{(i,j) \in M_k} P_{\max}(i,j) \tag{9}$$

where $d_{P\text{-}P}$ denotes the distance between the atoms P; $N_{bins}$ is the total number of distance bins (38 here), $M_k$ is a collection of nucleotide pairs $(i,j)$ (from $S$), for which the maximum probability of $d_{P\text{-}P}$, (i.e., $P_{\max}(i,j)$), belongs to the $k^{th}$ distance bin.

3. *std*, the average standard deviations of the probability values for all nucleotide pairs.
4. *prop*, the proportion of nucleotide pairs with $P(d_{P\text{-}P} < 40\,\text{Å}) > 0.45$.

The RMSD is estimated based on linear regression over the above variables using hundreds of randomly selected RNAs from the training set.

$$eRMSD = 0.64 \times pRMSD - 189.43 \times std - 4.01 \times mp - 1.06 \times prop + 15.2 \tag{10}$$

### Statistics & reproducibility

No statistical method was used to predetermine sample size. No data were excluded from the analyses. The experiments were not randomized. The Investigators were not blinded to allocation during experiments and outcome assessment.

### Reporting summary

Further information on research design is available in the Nature Portfolio Reporting Summary linked to this article.

## Data availability

The training sets, the set of 20 RNA-Puzzles RNAs, and the set of 30 independent RNAs can be downloaded from Zenodo[51] and our website (https://yanglab.qd.sdu.edu.cn/trRosettaRNA/). The RNAs from blind tests of CASP15 and RNA-Puzzles can be downloaded from https://predictioncenter.org/casp15/results.cgi?tr_type=rna and https://www.rnapuzzles.org/results/, respectively. The PDB entries mentioned in this study (3IVK, 5KH8, 5LYS, 6D89, 7D7V, 8DP3, 8GXC, and 8HB8) were obtained by four-digit accession codes in the Protein Data Bank repository (https://www.rcsb.org/). The sequence databases of NCBI's nt, Rfam, and RNAcentral used to generate MSA in this study can be downloaded from https://www.ncbi.nlm.nih.gov/nucleotide, https://rfam.org/, and https://rnacentral.org/, respectively. The source data underlying Tables 1, S7 and Figs. 4–6, S2, S3, S5, S6, S10, S11 are provided in the Source Data file. Source data are provided with this paper.

## Code availability

The trRosettaRNA server and the standalone package are available at Zenodo[51] and our website (https://yanglab.qd.sdu.edu.cn/trRosettaRNA/).

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

## Acknowledgements

This work is supported in part by the National Natural Science Foundation of China (NSFC T2225007 to J.Y., T2222012 to Z.P., and 61932018 to F.Z.), and the Foundation for Innovative Research Groups of State Key Laboratory of Microbial Technology (WZCX2021-03 to J.Y.).

## Author contributions

J.Y. conceptualized and administered the study. W.W. designed and implemented the network. C.F. and L.Y. implemented the energy minimization. Z.P. and F.Z. co-supervised the study. R.H., Z.W., Z.D., and H.W. prepared the training data. All authors revised and approved the final draft of the manuscript.

## Competing interests

The authors declare no competing interests.
