## [Peer Review File · Nature Communications]

trRosettaRNA: automated prediction of RNA 3D structure with transformer networkEditorial Note: This manuscript has been previously reviewed at another journal that is not operating a transparent peer review scheme. This document only contains reviewer comments and rebuttal letters for versions considered at *Nature Communications*.

Reviewer #1 (Remarks to the Author):

I am pleased to see that the revision has significantly improved the accuracy of the description of trRosettaRNA's performance, by acknowledging that challenges remain in predicting RNA tertiary structures with accuracy. The novelty and significance of the method presented in this work make it a valuable contribution to the field, but it is important to avoid exaggerating its performance, as this could be misleading and potentially frustrating for other researchers.

1, It's important to acknowledge that the CASP15 targets are more challenging than the benchmarks, as the CASP15 targets are real cases! Therefore, it's crucial to highlight the limitations of the benchmark in the conclusion to provide a comprehensive understanding of the results.

2, I recommend that the data splitting be done in a simple and easy-to-understand way, such as by using a sequence identity cutoff of 30% or lower. Ideally, there should be no similar sequence between the training and test datasets, but I know this is not practical. If the authors cannot repeat the work with this commonly-used criterion, they should at least provide a warning that the performance of trRosettaRNA for novel sequences or structures may be much lower.

3, Finally, it is necessary to compare trRosettaRNA with RhoFold (or AIchemy_RNA), another deep learning-based tool in CASP15, in terms of prediction accuracy.

Reviewer #2 (Remarks to the Author):

The authors have made a real effort to comply with the suggestions and comments made. The document is now much more balanced and presents a constructive perspective, not just a progress report. There is still some overuse of the word "outperform", but I leave that to the authors. I have only minor comments to make. In the main document, on page 8 (bottom, first line in red), I do not understand what the authors mean by "hinder" helices. In the rebuttal, response to reviewer 1, comment 4: the N3(1) atoms of the bases have been used. This is ambiguous; N3 and N1 in purines and pyrimidines have no structural similarity. I hope the authors mean N9 for purines and N1 for pyrimidines.

Reviewer #3 (Remarks to the Author):

Yang and colleagues have updated their paper on trRosetta-RNA in response to reviews. The paper's overall tone is much more muted and scholarly than prior versions, and the emphasis on results of blind challenges has improved the paper. I have some suggestions for minor revisions:

- The authors have not yet provided a complete explanation of the difference in performance between their internal benchmarks and CASP. The new figures and results on differences in secondary structure accuracy are welcome and provide a partial explanation. But the authors, in their response to Reviewer #3, state that the benchmark test sets have "relatively simpler topologies" and "one-third of the RNAs ... exhibit similar topologies to solved RNAs (TM-score_{RNA}>0.6)". This seems like a promising explanation, but is not backed up by appropriate data. The authors should provide graphs of TM-score_{RNA} to prior available structures compared to RMSD and TM-score of their model output to the actual structure, from their benchmark and for CASP/RNA-Puzzles molecules. If the explanation is correct, there should be statistically significant correlations; and the TM-score_{RNA} values to prior structural templates for CASP15 targets should be notably lower than in their benchmark. TM-score thresholds of 0.45 and 0.6 (homology match and very good homology match) could be shown on these diagrams. These comparisons would fit well in Fig. 2 which investigate other explanations of what makes targets hard for trRosetta-RNA.

- The way that the authors evaluate their performance in different test sets remains variable — it still seems suspicious, as if the authors are finding the metric for each test set that makes their

method best. The main example of this that I see is that in the RNA-Puzzles targets section, the comparison is based on "model 1", while in CASP the comparisons are made across best of 5. Perhaps both kinds of comparisons should be made for each of these two sets as well as the benchmark set and the latest RNA-Puzzle (blind) test.

- Similarly, the number for average trRosetta-RNA RMSD for the benchmark RNA's seems to change every time it is mentioned. It is 4.9 Å on p. 3; 5.0 Å on p. 4; 5.6 Å on p. 9; and 5.5 Å on p. 11. This kind of inconsistency makes it hard to trust the other numbers in the manuscript. My recommendation would be to provide a table (at least in supplemental as an Excel file) that encapsulates all statistics described in the manuscript, with a separate row for each benchmark molecule, and a final row with average RMSD, so that the authors as well as readers can check for cross-consistency.

- "The synthetic RNAs do not have any ... similar structures to the existing RNAs" (p. 8, line 229) — this is not true. Most of these RNA's were explicitly designed to include motifs like the HIV DIS kissing loops, aptamers and four-way junctions. For example, the kissing loop in Fig. S10b matches NMR structures of the HIV DIS loop (see the papers describing the designs of these nanostructure). If these structural 'submotifs' are not easily detectable by the trRosetta-RNA model, it would be good to provide some explanation of why they are missed.

Response to Reviewers

Dear Reviewers,

We very much appreciate the valuable comments and suggestions from the reviewers. We have studied all comments carefully and made a careful revision on the original manuscript. We have made necessary adjustments to tone down any overstated claims and have included more objective discussions regarding the limitations of trRosettaRNA (e.g., in the DISCUSSION section). In addition, three figures (current Figs 2, 3, 4) from blind tests are moved from the supporting information to the main text, while moving the original Figs 2, 3 of the independent test set to the supporting information. Two tables (Tables 1 and 2) are added in the main text.

All revised portions are highlighted in red font in the revised manuscript. We also include detailed point-by-point replies to the comments from the reviewers below.

Sincerely,

Jianyi Yang (on behalf of all authors)

Professor, Shandong University

Response to Reviewer #1

COMMENT 1: *It's important to acknowledge that the CASP15 targets are more challenging than the benchmarks, as the CASP15 targets are real cases! Therefore, it's crucial to highlight the limitations of the benchmark in the conclusion to provide a comprehensive understanding of the results.*

REPLY: We are grateful to the reviewer for the valuable comments and suggestions. In response, we have incorporated a discussion regarding the intrinsic distinctions between the targets in benchmarks and those in the CASP15 blind tests. These distinctions encompass two key facets: firstly, the quality of predicted secondary structures; secondly, the resemblance to RNA structures that have been previously solved. Owing to these disparities, potential variations in modeling complexity between benchmark targets and CASP15 blind test targets may arise. For a more comprehensive discussion, please refer to the DISCUSSION section on page 16.

COMMENT 2: *I recommend that the data splitting be done in a simple and easy-to-understand way, such as by using a sequence identity cutoff of 30% or lower. Ideally, there should be no similar sequence between the training and test datasets, but I know this is not practical. If the authors cannot repeat the work with this commonly-used criterion, they should at least provide a warning that the performance of trRosettaRNA for novel sequences or structures may be much lower.*

REPLY: We extend our gratitude to the reviewer for the insightful suggestion. In response, we have included a comprehensive discussion addressing the limitations inherent in trRosettaRNA, which involves its constrained performance when confronted with novel structures. Nonetheless, we maintain our belief in the potential of deep learning to enhance RNA structure prediction by combining it with established conventional methodologies. For a more detailed exposition, please refer to the last paragraph within the DISCUSSION section on page 16.

COMMENT 3: *Finally, it is necessary to compare trRosettaRNA with RhoFold (or Alchemy_RNA), another deep learning-based tool in CASP15, in terms of prediction accuracy.*

REPLY: We express our sincere appreciation to the reviewer for the perceptive suggestion. In response, we have added the comparison with RhoFold immediately after our discussion comparing DeepFoldRNA and RoseTTAFoldNA. trRosettaRNA is competitive to RhoFold, which harnesses the capabilities of an RNA language model. This assertion is supported by evaluations conducted across both benchmark datasets and blind test scenarios. Please refer to the latter part of page 12 for more details.

Response to Reviewer #2

COMMENT 1: *The authors have made a real effort to comply with the suggestions and comments made. The document is now much more balanced and presents a constructive perspective, not just a progress report. There is still some overuse of the word "outperform", but I leave that to the authors.*

REPLY: We extend our gratitude to the reviewer for the positive feedback and insightful suggestions. We have tried our best to refine any overstated claims to make the manuscript more objective and balanced. In addition, to reflect the performance of our method more objectively, three figures (current Figs 2, 3, 4) from blind tests are moved from the supporting information to the main text, while moving the original Figs 2, 3 of the independent test set to the supporting information.

COMMENT 2: *In the main document, on page 8 (bottom, first line in red), I do not understand what the authors mean by "hinder" helices.*

REPLY: We appreciate the reviewer for the valuable comments. To clarify, the term intended should be "helix hinge". We apologize for any confusion caused by this mistake. We have thoroughly reviewed and corrected all instances of inaccurately typed phrases to ensure accuracy and clarity throughout the document.

COMMENT 3: *In the rebuttal, response to reviewer 1, comment 4: the N3(1) atoms of the bases have been used. This is ambiguous; N3 and N1 in purines and pyrimidines have no structural similarity. I hope the authors mean N9 for purines and N1 for pyrimidines.*

REPLY: We appreciate the reviewer for the valuable comments. Regarding the mention of N3(1) atoms in our response to the reviewer's comment, we want to clarify that the term "N3(1)" was intended to distinguish between the N3 atom in pyrimidines and the N1 atom in purines. The distinction is crucial as these specific atoms participate in hydrogen bonding interactions, specifically in Watson-Crick base pairs, such as A-U and G-C pairs. We apologize for any confusion and appreciate the opportunity to clarify this aspect.

Response to Reviewer #3

COMMENT 1: *The authors have not yet provided a complete explanation of the difference in performance between their internal benchmarks and CASP. The new figures and results on differences in secondary structure accuracy are welcome and provide a partial explanation. But the authors, in their response to Reviewer #3, state that the benchmark test sets have “relatively simpler topologies” and “one-third of the RNAs ... exhibit similar topologies to solved RNAs ($TM\text{-score_RNA} > 0.6$)”. This seems like a promising explanation, but is not backed up by appropriate data. The authors should provide graphs of $TM\text{-score_RNA}$ to prior available structures compared to RMSD and $TM\text{-score}$ of their model output to the actual structure, from their benchmark and for CASP/RNA-Puzzles molecules. If the explanation is correct, there should be statistically significant correlations; and the $TM\text{-score_RNA}$ values to prior structural templates for CASP15 targets should be notably lower than in their benchmark. $TM\text{-score}$ thresholds of 0.45 and 0.6 (homology match and very good homology match) could be shown on these diagrams. These comparisons would fit well in Fig. 2 which investigate other explanations of what makes targets hard for trRosetta-RNA.*

REPLY: We appreciate the reviewer for the valuable comments. In response, we have incorporated a new subplot (Fig. S2D) within the original Figure 2 (now Fig. S2), which illustrates the correlation between the model's RMSD and the maximum $TM\text{-score}_{RNA}$ to prior RNA structures. A corresponding discussion has been included in the highlighted paragraph on page 6.

Given that the original Figure 2 is intended to showcase the performance on the first benchmark test dataset consisting of 30 non-redundant RNAs, the aforementioned subplot exclusively pertains to these 30 RNAs to ensure the coherence of the manuscript's overall structure. For a comprehensive comparison across all utilized datasets, please refer to Figure S15, which involves the 30 independent RNAs, the 20 previous RNA-Puzzles targets, the 12 CASP15 RNAs, and the 3 latest RNA-Puzzles targets. As expected, **the structural similarity to prior structural templates for CASP15 targets (average: 0.369) is significantly lower than the RNAs in the benchmark (average: 0.695)**. Furthermore, when considering all the tested RNAs, the PCC between the model RMSD and the $TM\text{-score}_{RNA}$ values to prior structural templates is -0.486. These data can serve as an explanation of the difference in performance between benchmarks and CASP15 blind tests for our data-driven method, trRosettaRNA. For more details, please refer to the DISCUSSION section on page 16.

COMMENT 2: *The way that the authors evaluate their performance in different test sets remains variable — it still seems suspicious, as if the authors are finding the metric for each test set that makes their method best. The main example of this that I see is that in the RNA-Puzzles targets section, the comparison is based on “model 1”, while in CASP the comparisons are made across best of 5. Perhaps both kinds of comparisons should be made for each of these two sets as well as the benchmark set and the latest RNA-Puzzle (blind) test.*

REPLY: We appreciate the reviewer for the valuable comments and suggestions. We have included both the top 1 and top 5 RMSDs within the RNA-Puzzles, CASP15, and the latest RNA-Puzzles targets sections (the last sentence on page 6, the middle of page 8, the middle of page 10, Tables 2,

S3 and S5, as well as Figure 4).

For the comparison with traditional methods (SimRNA and RNAComposer) and other deep learning-based methods (DeepFoldRNA, RoseTTAFoldNA and RhoFold), we opted to solely evaluate the “model 1” to ensure uniformity across all methodologies, as some of these methods only produce a single model per target (e.g., RNAComposer, RoseTTAFoldNA and RhoFold). For clarity, we have explicitly articulated these particulars in the corresponding sections of the manuscript. Please refer to the end of page 3 and the first paragraph on page 11.

COMMENT 3: *Similarly, the number for average trRosetta-RNA RMSD for the benchmark RNA's seems to change every time it is mentioned. It is 4.9 Å on p. 3; 5.0 Å on p. 4; 5.6 Å on p. 9; and 5.5 Å on p. 11. This kind of inconsistency makes it hard to trust the other numbers in the manuscript. My recommendation would be to provide a table (at least in supplemental as an Excel file) that encapsulates all statistics described in the manuscript, with a separate row for each benchmark molecule, and a final row with average RMSD, so that the authors as well as readers can check for cross-consistency.*

REPLY: We appreciate the reviewer for the valuable comments and suggestions. We sincerely apologize for any confusion caused by the inconsistent numbering, which may have led to misunderstandings. To provide clarity, allow us to elucidate the specific contexts: the reference to "4.9 Å on p. 3" refers to the average RMSD calculated across all 39 RNAs; "5.0 Å on p. 4" refers to the average RMSD assessed solely on the 37 RNAs employed for comparison with SimRNA and RNAComposer; and "5.6 Å on p. 9" corresponds to the average RMSD computed for the combined set comprising 39 RNAs and 20 RNA-Puzzles targets.

To mitigate any confusion, we have revised the statements in question, enhancing their clarity and coherence. For precise details, please refer to the middle of page 11 and the middle of page 16. Notably, these numerical values have been modified in the revised manuscript, reflecting our decision to filter the original set of 39 RNAs to 30 by excluding any duplicates which also present in the 20 RNA-Puzzles targets.

We have diligently verified the consistency of statistical information within the revised manuscript and provided a table in supplemental materials (the first sheet in “Source Data.xlsx”), storing the RMSD value for all the benchmark RNAs.

COMMENT 4: *“The synthetic RNAs do not have any ... similar structures to the existing RNAs” (p. 8, line 229) — this is not true. Most of these RNA's were explicitly designed to include motifs like the HIV DIS kissing loops, aptamers and four-way junctions. For example, the kissing loop in Fig. S10b matches NMR structures of the HIV DIS loop (see the papers describing the designs of these nanostructure). If these structural ‘submotifs’ are not easily detectable by the trRosetta-RNA model, it would be good to provide some explanation of why they are missed.*

REPLY: We appreciate the reviewer for the valuable comments and suggestions. We apologize for any inaccuracies in our previous statements. It's worth noting that no existing solved RNA structures demonstrate evident **global** homology with the synthetic RNAs in CASP15. Their

maximum $TM\text{-score}_{RNA}$ stands at approximately 0.3. To rectify any potential misunderstanding, we have revised the corresponding sentence, emphasizing the "**global**" aspect of the homology comparison. Please refer to the first paragraph on page 9.

For the specific motifs, regarding R1138, our investigations have indicated that trRosettaRNA can indeed achieve accurate predictions for the kissing loop's structure when the focus is exclusively on this motif (Figs. S11A and B). However, when confronted with the task of folding the entire tertiary structure, the precision of modeling for this specific kissing loop diminishes, owing to intricate constraints imposed by other concurrent motifs (Fig. S11C). Notably, the Alchemy_RNA2 paper disclosed their utilization of a **motif-based** approach and **manually identifying kissing loops** within these synthetic RNAs ¹. Thus we conclude that it is still difficult to accurately model such complicated synthetic RNAs in a global and automated manner. Please refer to the discussion in the middle of page 9.

1. Ke, C., Yaoqi, Z., Sheng, W. & Peng, X. RNA tertiary structure modeling with BRiQ potential in CASP15. *bioRxiv*, 2023.2005.2026.542548 (2023).

Reviewer #3 (Remarks to the Author):

The authors have addressed my suggestions satisfactorily.